# Design and Synthesis of A PD-1 Binding Peptide and Evaluation of Its Anti-Tumor Activity

**DOI:** 10.3390/ijms20030572

**Published:** 2019-01-29

**Authors:** Abdul Baset Abbas, Bingjing Lin, Chen Liu, Arwa Morshed, Jialiang Hu, Hanmei Xu

**Affiliations:** 1The Engineering Research Center of Synthetic Polypeptide Drug Discovery and Evaluation of Jiangsu Province, China Pharmaceutical University, Nanjing 210009, China; abduabbas04@gmail.com (A.B.A.); 1621030498@stu.cpu.edu.cn (B.L.); hsdalan@163.com (C.L.); arw34129@gmail.com (A.M.); 2Department of Medical Microbiology, Faculty of Sciences, Ibb University, Ibb City 70270, Yemen; 3Nanjing Anji Biotechnology Co. Ltd., Nanjing 210046, China

**Keywords:** T cells, PD-1, PD-L1, FITC-YT-16 peptide, tumor, cytokines

## Abstract

Immune-checkpoint blockades, suchas PD-1 monoclonal antibodies, have shown new promising avenues to treat cancers. Failure responsesof many cancer patients to these agents have led to a massive need for alternative strategies to optimize tumor immunotherapy. Currently, new therapeutic developments involve peptide blocking strategies, as they have high stability and low immunogenicity. Here, we have designed and synthesized a new peptide FITC-YT-16 to target PD-1. We have studied FITC-YT-16 by various experiments, including Molecular Operating Environment MOE modeling, purification testing by HPLC and LC mass, peptide/PD-1 conjugation and affinity by microscale thermophoresis (MST), and T cell immune-fluorescence imaging by fluorescence microscopy and flow cytometry. The peptide was tested for its ability to enhanceT cell activity against tumor cell lines, including TE-13, A549, and MDA-MB-231. Lastly, we assessed T cell cytotoxicity under peptide treatment. YT-16–PD-1 interaction showed a high binding affinity as a low energy complex that was confirmed by MOE. Furthermore, the peptide purity and molecular weights were 90.96% and 2344.66, respectively. MST revealed that FITC-YT-16 interacted with PD-1 at a K_d_ value of 17.8 ± 2.6 nM. T cell imaging and flow cytometry revealed high affinity of FITC-YT-16 to PD-1. Interestingly, FITC-YT-16 efficiently blocked PD-1 signaling pathways and promoted T cell inflammatory responses by elevating IL-2 and INF-γ levels. Moreover, FITC-YT-16 has the ability to activate T cell cytotoxicity. Therefore, FITC-YT-16 significantly enhanced T cell anti-tumor activity by blocking PD-1–PD-L1 interactions.

## 1. Introduction

Cancer is a major cause of public health problems worldwide. Its incidence is associated with an increase in annual mortality [1,2,3,4]. T cells have potential to be stimulated by tumor cells, thus triggering immune responses. They are activated through the connection of the T cell receptor (TCR) with major histocompatibility complex (MHC) in antigen presenting cells (APC) and interaction of the co-stimulatory molecules. Unfortunately, tumors not only avoid the immune system by down regulating both co-stimulatory molecules and MHC, but also upregulate co-inhibitory molecules [5,6]. Expression and engagement of PD-1 and CTLA-4 with their ligands are associated with tumorigenesis prognosis and massiveness [7]. PD-1 is an immune checkpoint molecule expressed on T cells. It is a member of the CD28/B7/CTLA-4 family and is encoded by PD-1 gene placed on chromosome number 2 at q37 band [5,8]. PD-1 counter-receptors PD-L1 and PD-L2 convey potent T cells co-inhibitory signals [9]. In cancer, the upregulation of PD-L1 leads to down-regulation of T cell activity and may assist tumor immune avoidance [10]. It was reported that cancer cells mediate T cells polarization through PD-L1 (CD274) to down-regulate immunosurveillance and enhance tumor proliferation by releasing of TGF-β, IL-10, and other regulatory cytokines [11,12]. Engagement of tumor PD-L1 withPD-1on T cells delivers a signal that attenuatesP13K/Akt, RAS/MEK/ERK, PKC-θ, and glycolysis activation loop phosphorylation. Moreover, PD-L1 binding to PD-1 also contributes to ligand-induced TCR down-modulation during antigen presentation to naive T cells [13,14,15].

Blocking of the interaction between PD-L1 or PD-L2 expressed on tumor cells and PD-1 on the T-cells has potential application in regulating the immune response against cancers (Figure 1). Obviously, in order to achieve optimum activation, the interactions of PD-1/PD-L1 orPD-1/PD-L2 needs to be blocked. Therapeutic antibodies against either PD-1 or PD-L1 are widely used to repair handicapped T cells’ ability to proliferate, secrete cytokines, and kill tumor cells. Therefore, implementation of therapeutic antibody blockade of either PD-L1 or PD-1 leads to increased antitumor immunity [16,17,18]. Various monoclonal antibodies (mAb) have been approved for this aim, such as Cemiplimab, Nivolumab, Pembrolizumab, Atezolizumab, Avelumab, and Durvalumab. These antibodies block the PD-1/PD-L1 or PD-1/PD-L2 interactions to treat several types of tumors [19,20,21,22,23]. However, monoclonal antibodies have a number of potential disadvantages, which include high production costs and immune-related adverse events (irAEs) that cause cytotoxicity, affecting normal tissues. Indeed, the massive immunogenicity of mAbs virtually enhances autoimmune reactions [7,8,13].

The field of medical peptides and small molecules is expanding and could be one of the future medications in the treatment of diseases. Many studies have been concerned with the design, synthesis, extraction, and evaluation of the effectiveness of such molecules. In recent years, peptide therapy has become one of the promise candidates for drug development in cancer treatment and other diseases [24,25,26,27,28,29,30,31,32]. Jolene L. Lau and Michael K. Dunn said that peptides represent one of the greatest areas of pharmaceutical development, particularly for tumor treatment, metabolic disease, and cardiovascular disease [33].

Peptides are a unique type of pharmaceutical compound and their molecular weights are between those of proteins and small molecules [33]. Peptides have a number of properties, such as exquisite sequence selectivity, noticeable efficacy, and low toxicity. Moreover, a large number of peptides are currently available and identified, all of which positively participate in a diverse range of physiological mechanisms, with roles as growth factors, neurotransmitters, ion channel ligands, hormones, antimicrobials, and immunomodulators [34,35,36,37]. A novel peptide represents an excellent starting point in drug development. In addition, peptide specificity and selectivity to their target receptors, their sequence flexibility, and conjugation possibilities make them good drug candidates [38,39]. Compared to large biomolecules, the peptide can penetrate deep tissue in the body. Furthermore, peptides have minor immunogenicity, minimal toxicity, greater efficacy, and are relatively cheap, easy to produce, and more easily stored than antibodies and recombinant proteins [38,39,40,41,42,43,44]. Small molecules and peptides were also reported to target PD1 or PD-L1 as antagonists and they showed enhanced anti-tumor activity and less immune-related side effects [8,21,42]. 

In this study, we have designed a novel YT-16 sequence that targets PD-1 by computational peptide design method [45] and prepared FITC-YT-16. To determine FITC-YT-16 peptide and PD-1 interactions, a docking analysis was performed by Molecular Operating Environment (MOE 2009). FITC-labeled peptide was produced by a chemical method on a solid-phase peptide synthesis (SPPS) [46]. Its purity and molecular weight were evaluated by HPLC and mass spectrometry. Also, we have demonstrated the interactions of FITC-YT-16/PD-1 by microscale thermophoresis (MST) analysis, cellular imaging, and flow cytometry analysis. Furthermore, we confirmed that FITC-YT-16 can enhance T cell anti-tumor activity by inducing greater cytokine secretion and improved cytotoxicity by in vitro experiments. 

## 2. Results

### 2.1. Design of YT-16 and Its Interactions With PD-1

Based on the crystal structure of PD-1 and PD-L1, the Asn (68), Gln (75), Thr (76), Lys (78), Ile (126), and Glu (136) in the PD-1 sequence were found to play a key role in PD-1/PD-L1 interactions. So, we designed a virtual peptide library that can interact with PD-1 with potential affinity. These peptides were designed based on the corresponding PD-L1 interacting fragment and the property of the five accepting amino acids in the PD-1 molecule. The Asn (68), Gln (75), Thr (76), Lys (78), Ile (126), and Glu (136) in PD-1 were selected and defined as accepting residues. Then, MOE 2009 was used to analyze docking of the designed peptides in Table 1 with PD-1. The affinity between PD-1 and the designed peptide is shown in Table 1. The peptide YT-16 (YRCMISYGGADYKCIT(C-C)) had the highest affinity. The combination diagram of YT-16 and PD-1is shown in Figure 2. From Figure 2, it can be seen that the N terminal part of YT-16 (YRCMISY) interacted with residues Ile (126) and Glu (136) of PD-1, whereas the C terminal part of YT-16 (ADYKCIT) interacted with residues Asn (68), Gln (75), Thr (76), and Lys (78) of PD-1. These simultaneous interactions benefit from the cyclic conformation of YT-16. If YT-16 is a linear peptide, the simultaneous interactions may not be fulfilled. YT-16 was synthesized for further evaluation.

### 2.2. Synthesis of FITC-YT-16

As mentioned above, peptide YT-16 (YRCMISYGGADYKCIT(C-C)) was designed with binding specificity to human PD-1 by the computational peptide design method [28]. YT-16 was synthesized by solid phase peptide synthesis using Fmoc-protected amino acids (GL Biochem Ltd, Shanghai, China), and it was conjugated at the N terminus with fluorescein isothiocyanate isomer (FITC). In order to avoid steric hindrance, an ε-aminocaproic acid residue (Acp) linker was used in between the YT-16 peptide and the FITC molecule (Figure 3).

### 2.3. HPLC and LC-Mass

After synthesis, the peptide was prepared by cleavage off the resin, precipitation, and HPLC purification. Then, FITC was conjugated to YT-16 via an Acp linker. HPLC detection of FITC-YT-16 purity was rated 90.96% as presented in Figure 4A. By electrospray ionization-mass spectrometry analysis (ESI-MS), the main signal for FITC-YT-16 was at 1171.35 ([M − 2H]^2−^). The molecular weight was calculated to be 2344.66 which is identical to the predicted molecular weight of FITC-YT-16 (Figure 4B).

### 2.4. FITC-YT-16/PD-1 Affinity

Microscale thermophoresis (MST) is a powerful technique used to determine the bimolecular interaction quantity. It depends on thermophoresis, the directed transfer of molecules in a temperature gradient, which is based on a variety of molecular properties, including charge, size, conformation, and hydration shell [47]. The experiment was performed on a Monolith NT.115 device with 233 nM FITC-YT-16 and various concentrations of PD-1 protein at 20% MST power, 20% LED power, and 25 °C in a premium capillary device at 25 °C. The concentrations of PD-1 protein were from 0.78 nM to 3188.75 nM. As shown in Figure 5A, the difference of the fluorescence signal of the ten capillaries was within 10%, indicating the sample loading amounts into the ten capillaries were similar. The thermophoresis curves in Figure 5B were smooth without serrations, indicating that there were no sample aggregations and the parameters set were in accordance with detection requirements. The MST measurements revealed that FITC-YT-16 interacted with PD-1 at a dissociation constant (K_d_) value of 17.8 ± 2.6 nM, as shown in Figure 5C.

### 2.5. PD-1 and PD-L1 Expression

Interactions between PD-1 on T cells and PD-L1 on tumor cells generate inhibitory signals to decrease T cell activity [48]. As the actions of PD-1 antagonist was based on the pre-requirement that it blocks PD-1/PD-L1 interactions, we detected the expression of PD-L1 on tumor cells and selected those PD-L1-expressing tumor cells to set up the in vitro experimental model and to evaluate the activity of our PD-1 peptide antagonist. It was found that esophageal squamous cell carcinoma (TE-13), lung carcinoma cell line (A549), and human breast adenocarcinoma (MDA-MB-231) highly express PD-L1 (Figure 6A). In addition, PD-1 was found to be abundantly expressed on activated T cells (Figure 6B). 

### 2.6. Fluorescence Imaging of FITC-YT-16 Binding to T Cells

Binding of FITC-YT-16 with PD-1 on T cell surfaces was first confirmed with detection under a fluorescence microscope. T cell membrane was stained with fluorescent Dil (red color) and FITC-YT-16 (green color). If FITC-YT-16 interacts with PD-1 on the T cell surface and stains the membrane green, merging with the red color of Dil yields a yellow color. T cells were activated by incubation of freshly isolated human T cells with human T-activator CD3/CD28 Dynabeads (Gibco, Life Technologies, Grand Island, NY, USA) for 48 h at 37°C. As shown in Figure 6, after staining activated T cell membrane with Dil (red) and with anti-PD-1 peptide (green) and examination under a fluorescence microscope, Dil stained the T cell membranes red and FITC-YT-16 stained T cells green. Their merge resulted in yellow, which confirmed that FITC-YT-16 interacted with PD-1 on the T cell surface (Figure 7, lower case). Furthermore, when we used Dil and FITC-YT-16 to stain unactivated T cells on which minimal PD-1 was expressed, Dil can still stain T cells whereas FITC-YT-16 was unable to stain unactivated T cells and their merge did not result in yellow. This confirmed that the green color of FITC-YT-16 stained activated T cells as a result of the specific binding of FITC-YT-16 with cell surface PD-1, rather than nonspecific binding of FITC-YT-16 with other cell membrane components (Figure 7, upper case).

### 2.7. Flow Cytometry Analysis of FITC-YT-16 Binding to T Cells

Binding of FITC-YT-16 with PD-1 on the surface of an activated T cell was also confirmed by flow cytometry analysis of the fluorescence signal after incubation with different concentrations of FITC-YT-16. As shown in Figure 8, activated T cells were incubated with 10, 100, or 1000 nM FITC-YT-16 for 2 h. Isotype antibody was used as a control and to define the gate for positive fluorescence signal. After incubation with FITC-YT-16, the rates of T cells with positive fluorescence signal were 0.243, 1.64, and 61.1%. This experiment confirmed that FITC-YT-16 interacts with PD-1 on T cells.

### 2.8. In Vitro Activity of FITC-YT-16 Binding

#### 2.8.1. Inhibition of T Cell Activity by PD-L1-Expressing Tumor Cells

Activated CD4 T cells can secret cytokines, e.g. IL-2 and IFN-γ, which stimulate the immune cells for proliferation and activation. Therefore, in the experiment presented in Figure 9 and Figure 10, we aim to evaluate the effect of blocking the interactions between PD-1 and PD-L1 by FITC-YT-16 on T cell activity, which is evaluated by measuring secreted IL-2 and IFN-γ levels in culture supernatant. In the experiment in Figure 9, we first tried to find the tumor cell to T cell ratio under which condition the levels of IL-2 and IFN-γ decreased significantly compared to the sample of T cells, and we did not have to culture a large quantity of tumor cells. Based on this tumor cell to T cell ratio, we evaluated the enhancing effect of FITC-YT-16 incubation on T cell activity by measuring IL-2 and IFN-γ levels in the culture supernatant (Figure 10).

Logically, the incubation of PD-L1-expressing tumor cells with T cells was accompanied by inhibition of T cell activity, e.g. inhibition of IL-2 and IFN-γ secretion by T cells. To evaluate the activity of T cells, we co-cultured TE-13, A549, and MDA-MB-231 cells that highly express PD-L1 (Figure 6) with T cells in different ratios as presented in Table 2. This was confirmed by an experiment in Figure 9. The ratio was tumor cell to T cell ratio. From Figure 9, co-culture of tumor cells with T cells decreased the levels of IL-2 and IFN-γ secreted by T cells for all three-tumor cell lines. This inhibition strengthened with the increase of tumor cell to T cell ratio. As presented in Figure 9A–C a tumor cell to T cell ratio of 4:1 showed a significant reduction of IL-2 levels, in which case a relatively small number of tumor cells were needed. However, the effect of a tumor cell to T cell ratio on INF-γ secretion was less significant than IL-2 (Figure 9D–F). A tumor cell to T cell ratio of 16:1 showed a significant reduction of both IL-2 and IFN-γ levels. These results indicated that tumor cell lines down-regulated T cell pro-inflammatory cytokine secretions significantly at a tumor cell to T cell ratio of 16:1. This ratio was used in the following FITC-YT-16 activity detection. For the samples with tumor cells (TE-13, A549 or MDA-MB-231) and without T cells, the levels of IL-2 and IFN-γ in cell culture were under the detection limits of the ELISA kits.

#### 2.8.2. Increased T Cell Cytokine Secretion by FITC-YT-16 Blockage of PD-1/PD-L1 Interactions

From the illustrated experiment in Figure 9, a tumor cell to T cell ratio of 16:1 was used to establish a platform to evaluate the effect of FITC-YT-16 incubation. These experiments showed that PD-1 signaling blockage by FITC-YT-16 resulted in significantly enhanced pro-inflammatory cytokines production as illustrated in Figure 10. IL-2 level was significantly increased in the culture systems which indicated that T cell activity was enhanced as shown in Figure 10A–C. This effect was confirmed by determination of significantly enhanced secretion of IFN-γ in the same culture systems as presented in Figure 10D–F. This suggested that T cell activity was recovered by blocking of PD-1/PD-L1 interaction by FITC-YT-16. Tumor cells with T cells in absence of FITC-YT-16 were used as a negative control and tumor cells with T cells in presence of a commercially available PD-1/PD-L1 inhibitor 3 (a cyclic peptide) were used as a positive control. 

#### 2.8.3. Lactate Dehydrogenase Release (LDH) Assay

The experiment in Figure 10 mainly evaluated the activation status of CD4^+^ T cells by detection of IL-2 and INF-γ levels in the culture supernatant. The test in Figure 11 mainly evaluated the enhanced cytotoxicity by CD8^+^ T cells by detection of lactate dehydrogenase (LDH) release in the culture supernatant. Activated CD8 T cells can recognize tumor cells and kill tumor cells by means of degranulation and Fas/FasL recognition. After the tumor cells were damaged, a cytoplasmic enzyme with name lactate dehydrogenase (LDH) is released from damaged cells into the supernatant and by measuring the levels of LDH, we can evaluate the cytotoxic effect of CD8 T cells. From Figure 11, FITC-YT-16 significantly enhanced the release of LDH to the culture medium. We noticed that blocking of T cell PD-1 enhanced T cell cytotoxicity, resulting in tumor cell recognition and killing. This indicated that PD-1 inhibition by FITC-YT-16 not only enhanced cytokine secretion, but also enhanced cellular cytotoxicity. In addition, we noticed that the cytotoxicity was dose-dependently increased with FITC-YT-16 concentration. The highest cytotoxicity was noticed with 16 µMFITC-YT-16. Tumor cells with T cells in absence of FITC-YT-16 were used as a negative control and tumor cells with T cells in presence of a commercially available PD-1/PD-L1 inhibitor 3 were used as a positive control.

## 3. Discussion

Engagement of PD-1 on T cells and PD-L1 on tumor cells transduces a signal that inhibits T cell cytolysis, cytokine production, and proliferation. Several lines of evidence suggest that PD-1 is a hot antitumor target on the surface of tumor-infiltrating T cells. High expression of tumor PD-L1 showed strong association with high tumor prognosis, suggesting that PD-1 is a key regulator of T cell immunosuppressive responses [49]. The PD-1 blocking strategy has been extensively reported. It showed T cell function recovery that proved the therapeutic importance of PD-1 targeting, however, most monoclonal antibodies against PD-1 show highly cytotoxic side effects [7,13]. According to available reports, peptides targeting the PD-1/PD-L1 interaction are an important and beneficial strategy for cancer treatment. The field of medical peptides may form the basis for a novel immunomodulatory molecule. Furthermore, a peptide is a feasible platform on which to create a specific PD-1/PD-L1 inhibitor [8,28,42]. 

A novel strategy to block the PD-1 pathway without side effects and high toxic effects and with lower cost is needed. Therefore, we hypothesized that the designation of new peptide blocking PD-1 could provide an effective therapeutic strategy. Here, we designed a PD-1 antagonist peptide YT-16 and prepared FITC-YT-16 by a solid phase peptide synthesis method. FITC-YT-16 was assessed by HPLC (90.96%) and mass spectrophotometer (2344.66). FITC-YT-16 and targeted PD-1 showed conjugation activity in MOE analysis and a high-affinity value of 17.8 ± 2.6 nM in MST analysis (Figure 5). The binding potential of a designed peptide to PD-1 was also confirmed by flow cytometry and T cell fluorescence imaging. 

In this project, three solid tumor cell lines TE-13, A549, and MDA-MB-231 with high expression of PD-L1 were individually co-cultured with T cells in presence of FITC-YT-16. These experiments confirmed that FITC-YT-16 is able to maintain T cell viability and promoted T cell inflammatory responses by elevating IL-2 and INF-γ secretion levels. Qiao Li. et al reported the Ar5Y_4 peptide, an anti-PD-1 peptide, acts to elevate IL-2 secretion levels, as did our peptide FITC-YT-16 [28]. On the other hand, Chang Hao, et al. demonstrated that ^D^PPA-1 peptide, a PD-L1 inhibitor, acts to inhibit PD-1/PD-L1 interaction [42]. Furthermore, the cytotoxic activity of T cells also increased remarkably in the LDH test. Our analysis showed that T cell activity was enhanced by FITC-YT-16 incubation which indicated that PD-1/PD-L1 interaction is an important factor to inhibit T cell anti-tumor responses and facilitate tumor proliferation. So, FITC-YT-16 blocking PD-1/PD-L1 interactions increased T activity and activated their cytolysis potential and these effects were dose-dependent. FITC-YT-16 peptide acts as a PD-1/PD-L1 blocker and enhanced T cell anti-tumor activity against several tumor cells including TE-13, A549, and MDA-MB-231. By MST assay, YT-16 has a K_d_ value of 17.8 nM for its target PD-1. The K_d_ value of Ar5Y_4 peptide for PD-1 is 1.38 µM [29] and the Kd value of DPPA-1 for PD-L1 is 0.51 µM [43]. It is worthwhile to make further chemical modifications of YT-16, trying to increase its affinity with the target molecule and, if possible, improve its in vivo stability.

In recent years, there has been a huge leap in the design, construction, efficacy, and marketing of peptides [50,51,52,53,54]. The anti-tumor activity of PD-1 antagonist peptides, such as YT-16, should not be limited by its single use. For example, the combination of anti-CTLA-4 and anti-PD-1 monoclonal antibodies enhances their anti-tumor activity, although with high immune-related adverse events (irAEs). This prompts us to construct a novel anti-CTLA-4/anti-PD-1 dual function peptide as a combination therapy [55]. Also, the specific blockage of the PD-1 pathway enhances a chimeric antigen receptor (CAR), leading to enhanced eradication of tumor cells [56]. Therefore, the YT-16 peptide may be used as an immune function booster in combination with CAR-T therapy. Overall, the obtained results demonstrated that the FITC-YT-16 peptide has high affinity to PD-1 and it can restore T cell cytokine secretion and cytotoxicity functions of T cells.

## 4. Materials and Methods

### 4.1. Ethics Statement

All samples were collected under a protocol approved by the Ethics Committee of China Pharmaceutical University (Permit Number: SYXK2012-0035, date of approval: 20 July 2012), following written informed consent.

### 4.2. YT-16 Peptide Designed and Synthesis

Based on the crystal structure of PD-1 and PD-L1, we have designed a new trigger anti-PD-1 peptide with the sequence YRCMISYGGADYKCIT(C-C) by a computational peptide design method [45]. Asn (68), Gln (75), Thr (76), Lys (78), Ile (126), and Glu (136) play a key role in the relationship of PD-1 and PD-L1. So, we designed the virtual peptide library that can interact with PD-1. Asn (68), Gln (75), Thr (76), Lys (78), Ile (126), and Glu (136) were selected. To determined YT-16/PD-1 affinity, molecular operating environment (MOE 2009) was used to perform a docking analysis. PD-1 sequence was obtained from Protein Data Bank (PDB ID: 3BIK). YT-16 peptide was modeled using the homology model method. The predicted antagonist sequence was inserted into the three-dimensional structures of PD-1 under activation of different active sites. The docking procedure predicted the complex structures of interactions between the YT-16 peptide and PD-1, and then the affinity and required energy complex were determined. In addition, all binding sites at the PD-1 structure’s core and outer region were detected. It was synthesized on a solid phase peptide synthesizer (GL Biochem Ltd, Shanghai, China) and the N-terminal of the peptide was labeled with fluorescein isothiocyanate (FITC) via an Acp linker.

### 4.3. Purification and Identification of Peptide by HPLC and LC Mass 

The FITC-YT-16 peptide was separated by reversed-phase liquid chromatography (HPLC) on a Waters system using a 4.6×250 mm, Kromasil 100-5 C18 column (Varian Prostar HPLC system, CA, USA). The mobile phase consisted of two solvents, A and B, run in gradient elution. Mobile phase A, was 0.1% trifluoroacetic in acetonitrile, while 0.1% trifluoroacetic acid in water constituted the phase B. The elution system was as follows: at 0–25 min, 70% of B; at 25–25.1 min, 45% of B and 25.1–30 min, 0% of B. The flow rate was 1.0 mL/min and a 5 µL sample was loaded onto the column. The separated components were detected at 220 nm. 

To determine the molecular mass of the FITC labeled peptide, the chromatographic system coupled to a Waters ZQ2000 mass spectrometer was used. The operating conditions were as follows: mode, ESI negative; nebulizer gas flow, 1.5 L/min; CDL, −20.0 v; CDL temperature, 250 °C; block temperature, 200 °C; probe bias, +4.5 kv. The peptide was lyophilized and stored at −20 °C until use and reconstituted in phosphate buffered saline (PBS) solution for in vitro experiments.

### 4.4. Microscale Thermophoresis (MST) Binding Assay

Microscale thermophoresis (MST) is a powerful technique used to determine the biomolecular interactions quantity depends on the thermal motion [57]. Therefore, MST is able to describe protein–peptide binding. Here, it was achieved with 233 nM FITC-YT-16 in 1 mM EDTA, 1 mM DTT, 10 mM sodium phosphate (pH 7.0), and 0.5 mM PMSF with PD-1 protein (Cloud-Clone Corp, USA) at different concentrations (3188.75 nM, 1594.38 nM, 797.19 nM, 398.59 nM, 199.30 nM, 24.91 nM, 6.23 nM, 3.11 nM, 1.56 nM, and 0.78 nM), at 20% MST power and 20% LED power in premium capillaries on a Monolith NT.115 device at 2–6 °C (NanoTemper Technologies, Munich, Germany). Analysis of MST affinity was carried out using MST analysis software (MO. Affinity Analysis, Munich, Germany). Monolith NT^TM^ capillary “Premium coated” (NanoTemper Technologies, Munich, Germany) were loaded with 4 µL sample. Then, tubes were installed to NanoTemper’s dedicator to test at 1480 nm.

### 4.5. T Cell Activity Assay

#### 4.5.1. PBMCs Isolation and T Cell Isolation, Activation, and Culturing

To verify if FITC-YT-16 exerts a functional effect on human T cells, we assessed peptide performance on freshly isolated T cells. Briefly, human peripheral blood mononuclear cells (PBMCs) were isolated from healthy donor’s blood by density gradient centrifugation using Ficoll-Paque PLUS (GE Healthcare, Little Chalfont, Buckinghamshire, UK)). In brief, blood was diluted at 1:1 with phosphate-buffered saline (PBS), layered on Ficoll-Paque and centrifuged at 2000× *g* for 20 min. PBMCs were re-suspended in phosphate-buffered saline. Isolated T cells were immediately enriched by magnetic microbeads positive selection (Miltenyi Biotec GmbH, Bergisch Gladbach, Germany). For stimulating and activating T cells, isolated T cells were cultured in 6 well-plate in RPMI-1640 medium with 10% fetal bovine serum (FBS), 100 IU/mL penicillin, and 100 µg/mL streptomycin (Gibco, Life Technologies, Grand Island, NY, USA). Then inoculated with human T-activator CD3/CD28 Dynabeads (Gibco, Life Technologies, Grand Island, NY, USA, USA) at a ratio of one bead to one T cell for 24 h, supplemented with 50 IU/mL of recombinant human IL-2 (Novus Biologicals, Littleton, CO, USA).

#### 4.5.2. Fluorescence Imaging

Fluorescence microscopy allowed the investigation of the peptide interactions with activated and unactivated T cells. FITC-labeled peptide proved to bind to PD-1 on the activated T cell surface. T cells plated in 24-well culture plate and then treated with 1 mL 1% bovine serum albumin (BSA) for 30 min at 37 °C. Next, the cells were incubated with 300 nM FITC-YT-16 peptide (final concentration) at 37 °C for 1 h in dark condition. After incubation, cells were washed twice with PBS and then stained by Dil dye for 10 min. Finally, T cells were collected, washed twice with PBS and detected the binding of FITC-YT-16 peptide to PD-1 on activated and unactivated T cells surface by fluorescence microscopy (OLYMPUS IX53, Tokyo, Japan). For detection of yellow and green fluorescence, the excitation wavelength is 490–495 nm and the emission wavelength is 520–530 nm. For detection of red fluorescence, the excitation wavelength is 549 nm and the emission wavelength is 565 nm.

#### 4.5.3. Flow Cytometry Assay

To evaluate conjugation and affinity potential of the new designed FITC-YT-16 peptide to a PD-1, we performed flow cytometry assay in comparison to IgG3 isotype as a negative control (Biolegend, San Diego, CA, USA). Briefly, 1 × 10^5^ T cells were co-cultured with different FITC-YT-16 peptide concentrations (0, 10, 100, and 1000 nM). Thereafter, cultures were incubated under humid conditions, 5% CO_2_ and 37 °C for 2 h. Later, cells were collected, washed and then investigated by MACS Quant flow cytometer (Miltenyi Biotec GmbH, Bergisch Gladbach, Germany). Results were analyzed by FlowJo software version 7.6.1 (FlowJo LLC, Ashland, USA).

#### 4.5.4. Tumor Cell Lines and Cell Culture 

Esophageal squamous cell carcinoma (TE-13), lung carcinoma cell line (A549), human breast adenocarcinoma (MDA-MB-231) were purchased from the Cell Bank of the Chinese Academy of Sciences (Shanghai, China). Tumor cells were cultured in Dulbecco’s modified Eagle medium (DMEM) containing 100 µg/mL streptomycin, 50 IU/ml penicillin, 2200 µg/mL NaHCO_3_ and 10% (*v*/*v*) fetal bovine serum (FBS) (Gibco, Life Technologies, Grand Island, NY, USA), and incubated at 5% CO_2_ in a humidified incubator at 37 °C. 

#### 4.5.5. Western Blot Analysis of Protein Expressions 

Western blot was performed as previously described [26]. It was performed to analyze the protein expression in cells. Tumor cells and T cells were collected and washed twice with ice-cold PBS. Cells were lysed in RIPA lysis buffer which contains PMSF in ratio of 99: 1 (*v*/*v*) (Wanleibio Co. Ltd, Liaoning, China) in ice for 30 min according to the manufacturer’s instructions. Then the supernatant was collected by centrifugation at 12,000× *g* for 15 min at 4 °C. The concentrated protein was separated on a 10% SDS-polyacrylamide gel, and the protein bands were transferred to polyvinylidene fluoride (PVDF) membrane. Thereafter, membranes were blocked by incubating with 5% skim milk in TBST buffer for 2 h and incubated with specific primary antibodies (Abcam, Cambridge, MA, USA) with gentle agitation overnight at 4 °C. After that, the membranes were washed three times with TBST buffer and incubated at room temperature for 1.5 h with horseradish peroxidase-conjugated secondary antibody (Wuhan Servicebio Technology Co. Ltd, Wuhan, China), and then the membranes were washed by TBST. Finally, protein bands were visualized with an enhanced chemiluminescence detection kit (ECL, Millipore Corporation, Billerica, MA, USA).

#### 4.5.6. Cytokine Production Test by ELISA

Production of cytokines is an important indicator to evaluate the function of T cells [28]. Concisely, targeted cells (tumor cells) cells co-cultured with effectors cells (T cells) at a different ratio as illustrated in Table 2. The cells were co-cultured in a 96-well round-bottomed plate (Thermo Scientific™) using 200 µL RPMI-1640 and incubated at 37 °C, 95% humidity and 5% CO_2_ for 24 h. Then, the test plate was centrifuged at 1500 rpm for 20 min. Afterward, the supernatants were collected and IFN-γ and IL-2 levels were measured at 450 nm by using ELISA kit (MultiSciences Lianke Biotech Co., Ltd. Hangzhou, China).

As presented in Table 2, we performed experiments to select the proper ratio of the tumor cell to T cell, and we analyzed the secreted cytokines. Tumor cells to T cells ratio (16:1) revealed a significant of cytokine secretion as showed in Figure 9. Thereafter it was selected to detect FITC-YT-16 efficacy. Briefly, targeted (Tumor) cells co-cultured with effectors (T) cells at ratio 16:1 in the presence of different FITC-YT-16 concentrations (1, 2, 4, 8, and 16 µM) on a 96-well round bottom plate and incubated 24 h at 37 °C in the presence of 5% CO_2_. The supernatant was collected by test plate centrifugation at 1500 rpm for 20min. ELISA plates (MultiSciences Lianke Biotech Co., Ltd. Hangzhou, China) were used to evaluate IL-2 and INF-γ levels. It was performed using a 10 µL PD-1/PD-L1 inhibitor 3 cyclic peptide (Selleck Chemicals, Houston, TX, USA) as a positive control and a mixture of the tumor to T cell (16:1 ratio) in absence of FITC-YT-16 as a negative control. Three replicates of each experimental condition were performed. GraphPad software (version 5.01; GraphPad Inc. La Jolla, CA, USA) was used to analyze the results.

#### 4.5.7. Lactate Dehydrogenase (LDH) Release Assay

Effect of FITC-YT-16 peptide on T cell cytotoxicity was detected by measuring the release of a cytoplasmic enzyme, lactate dehydrogenase (LDH), from damaged cells into the supernatant by using a colorimetric assay according to cytotoxicity detection kit (Cayman Chemical, Ann Arbor, MI, USA). TE-13 and T cells were plated on a 96-well round bottom plate. Concisely, 2 × 10^5^ T cells and 3.2 × 10^6^ TE-13 cells were cultured in a 96-well round-bottom plate in RPMI-1640 medium, treated with different concentrations of FITC-YT-16 (1, 2, 4, 8, and 16 µM) and incubated overnight at 37 °C in the presence of 5% CO_2_. After the incubation, the test plate was centrifuged at 1500 rpm for 20 min. The supernatant was transferred to a new 96-well assay plate and the LDH reaction solution was added to each well and incubated for 30 min at 37 °C before the absorbance was read at 490 nm. The incubated cells in medium without peptide served as a negative control and PD-1/PD-L1 inhibitor 3 cyclic peptide (Selleck Chemicals, Houston, TX, USA) as a positive control. Triplicate wells were averaged and the percentage of specific lysis was calculated by the following equation:
$$\%\;\mathrm{Cytotoxicity}=\left[\frac{\mathrm{Experimental}\;\mathrm{value}\;A490-\mathrm{Spontaneous}\;\mathrm{release}\;A490}{\mathrm{Maximum}\;\mathrm{release}\;A490-\mathrm{Spontaneous}\;\mathrm{release}\;A490}\right]*100$$


### 4.6. Statistical Analysis 

The data were analyzed by GraphPad software (version 5.01; GraphPad Inc. La Jolla, CA, USA). The results were expressed as means ± SD and *p*-values, 0.05 were considered to be statistically significant (* *p* < 0.05; ** *p* < 0.01; *** *p* < 0.001). 

## 5. Conclusions

In summary, we designed and synthesized a novel anti-PD-1 peptide. It has a high affinity to PD-1 and efficiently binds to PD-1-expressing activated T cells. It also efficiently enhanced the anti-tumor activity of activated T cells by enhancing cytokine secretion and their cytotoxic effect. Although it is too early to use FITC-YT-16 peptide as a strategy to treat tumors by blocking PD-1 pathway, it provides another example proving the concept that blocking PD-1/PD-L1 interactions by an antagonist peptide can enhance the anti-tumor activity of T cells, and this peptide may facilitate the evaluation of its combined use with an antagonist of other immune checkpoint molecules, such as CTLA-4, or even its use in combination with CAR-T cell strategy. Further modifications of this peptide are worthwhile to obtain a new peptide with higher affinity to its target, PD-1, and better bioavailability.

## Figures and Tables

**Figure 1 ijms-20-00572-f001:**
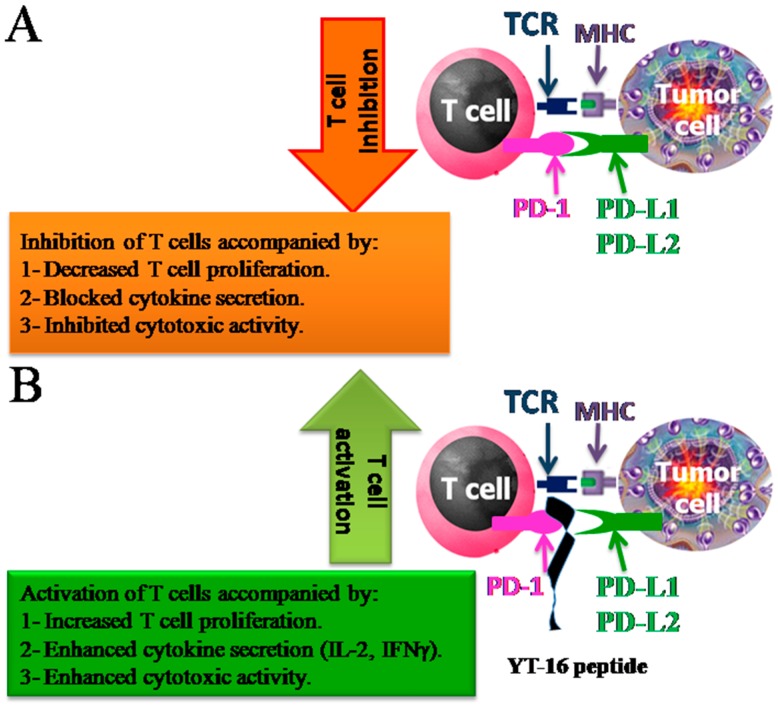
Schematic drawing of the interactions between PD-1 on T cells and PD-L1/PD-L2 on tumor cells. (**A**) PD-1/PD-L1 interaction in absence of YT-16 peptide accompanied by decreased T cell proliferation, blocked cytokine secretion and inhibited cytotoxic activity. (**B**) PD-1/PD-L1 interaction in presence of YT-16 accompanied by increased T cell proliferation, enhanced cytokine secretion such as (IL-2, IFN-γ) and enhanced cytotoxic activity. The functions of the interactions were also summarized.

**Figure 2 ijms-20-00572-f002:**
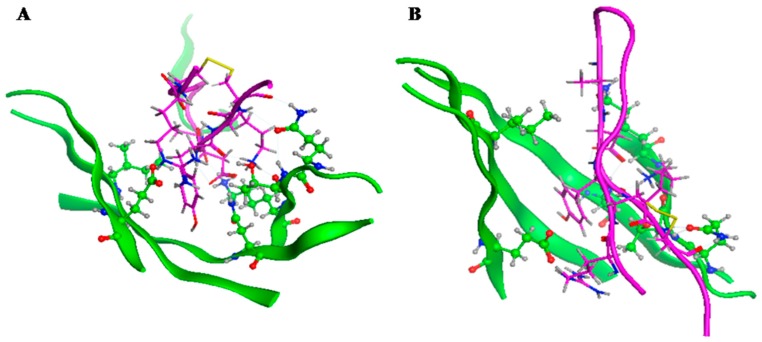
Combination diagram of YT-16 and PD-1. PD-1 is shown in green and YT-16 is in magenta. Panels **A** and **B** show the interactions seen from different angles.

**Figure 3 ijms-20-00572-f003:**
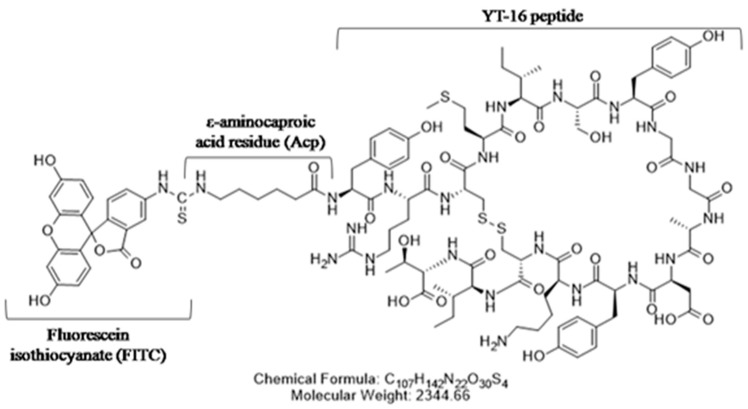
Chemical structure of FITC-YT-16. An Acp linker was used between FITC and YT-16.

**Figure 4 ijms-20-00572-f004:**
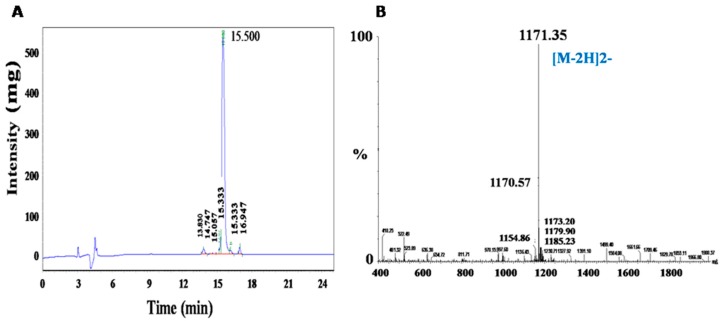
High-performance liquid chromatography (HPLC) and electrospray ionization-mass spectrometry (ESI-MS) confirmed the purity and identification of FITC-YT-16. (**A**) The purity of FITC-YT-16 was measured to be 90.96%. (**B**) ESI-MS measured the molecular weight to be 2344.66.

**Figure 5 ijms-20-00572-f005:**
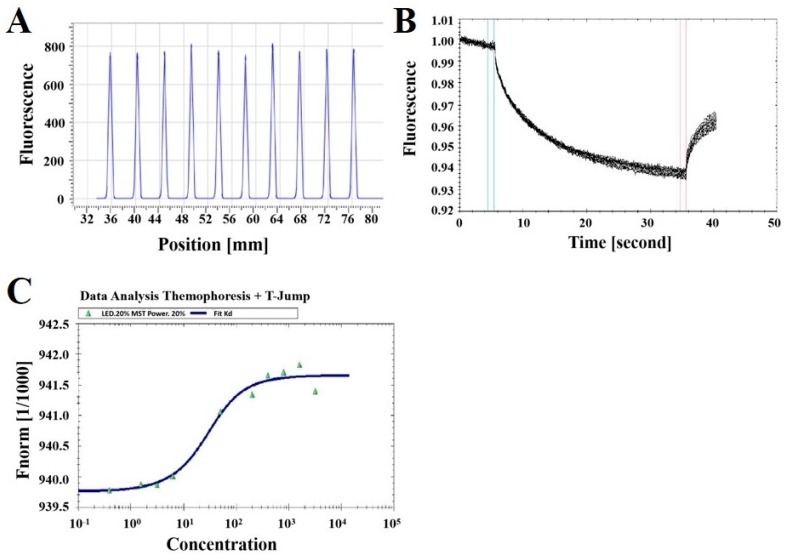
MST-determined binding affinity between FITC-YT-16 and PD-1 protein. (**A**) Fluorescence of the ten capillaries by capillary scan. (**B**) Thermophoresis curves. (**C**) Data Analysis of FITC-YT-16 and PD-1 thermophoresis. FITC-YT-16 bound with a high affinity to its target, human PD-1 (K_d_ = 17.8 ± 2.6 nM).

**Figure 6 ijms-20-00572-f006:**
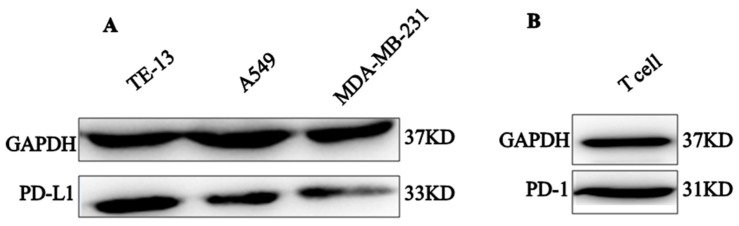
Western-blot analyses of PD-L1 and PD-1 expression. (**A**) Expression of PD-L1 on three human tumor cell lines. (**B**) Expression of PD-1 on activated T cells. GAPDH was used as a control.

**Figure 7 ijms-20-00572-f007:**
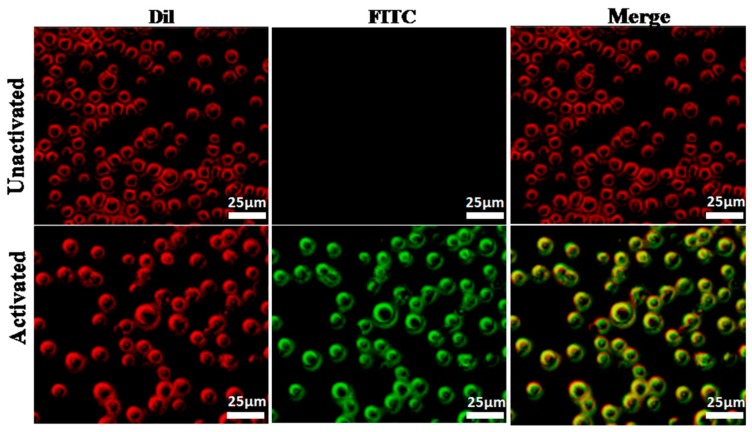
Binding of FITC-YT-16 peptide to activated and unactivated T cells. 300 nM FITC-YT-16 (PD-1 antagonist peptide) was incubated with activated or unactivated T cells. T cells were treated with Dil dye (red) and FITC-YT-16 peptide (green). The red color indicated T cell membrane, green color stated binding of FITC labeled peptide with PD-1 on the cell membrane, and the merge of two colors confirmed co-localization of FITC-YT-16 peptide and cell membrane (lower case). Scale bars represent 25 µm.

**Figure 8 ijms-20-00572-f008:**
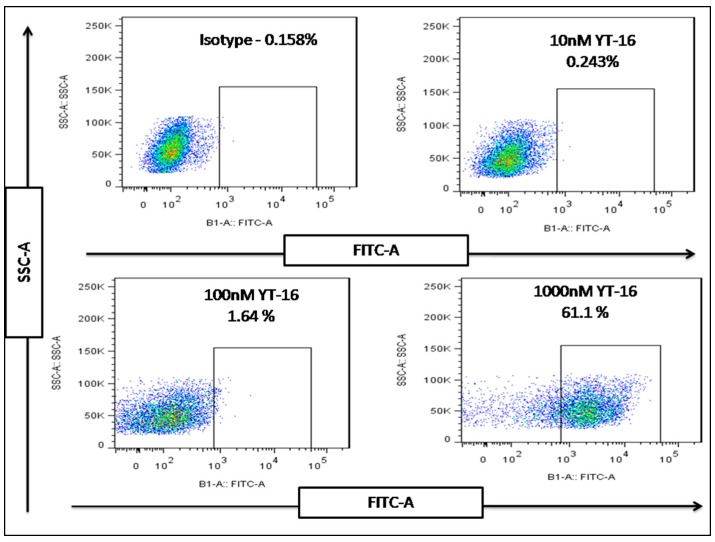
Binding of FITC/YT-16 peptide with PD-1 on T cells. T cells were incubated with 10, 100, and 1000 nM FITC/YT-16 peptide at 37 °C for 2 h. Binding of FITC/YT-16 with PD-1 was detected with flow cytometry technology. Isotype antibody was used as a control and to define the gate for positive fluorescence signal.

**Figure 9 ijms-20-00572-f009:**
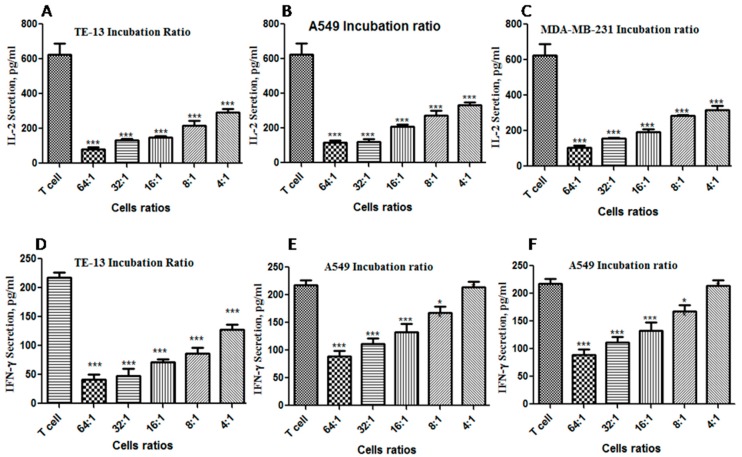
IL-2 and IFN-γ secretion by T cells in response to co-culture with different tumor cell to T cell ratio. (**A**) TE-13 co-culture with T cells reduced IL-2 secretion, (**B**) A549 co-culture with T cells significantly reduced IL-2 secretion, (**C**) MDA-MB-231co-culture with T cells also significantly reduced IL-2 secretion; T cell culture without tumor cells was used as a negative control. Moreover, the same tumor cell lines at same mixing ratios with T cells also showed a significant reduction of IFN-γ levels in the culture systems as illustrated in (**D**–**F**). There were three replicates for each experimental condition. * *p* < 0.05, ** *p* < 0.01 and *** *p* < 0.001, compared with the control group of T cells.

**Figure 10 ijms-20-00572-f010:**
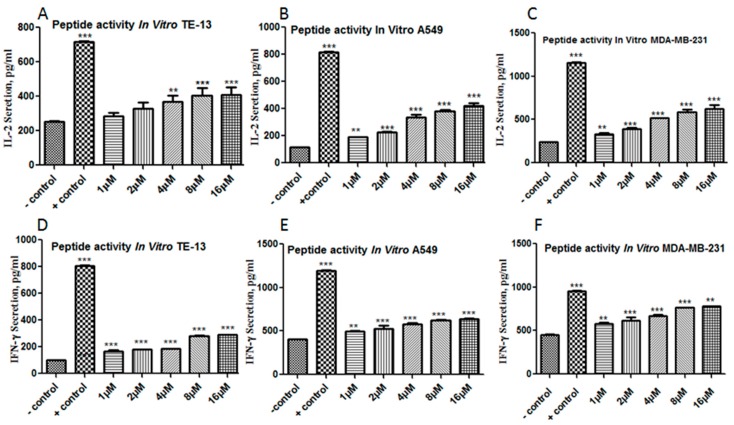
Enhanced T cells secretion of IL-2 and IFN-γ by FITC-YT-16 blockage of PD-1/PD-L1 interaction. FITC-YT-16 loaded T cells were incubated with three tumor cell lines at a tumor cell to T cell ratio of 16:1 with different FITC-YT-16 incubation concentrations (final concentrations of 1, 2, 4, 8, and 16 µM). Panels **A**, **B**, and **C** show significant elevated IL-2 levels with FITC-YT-16 incubation. This result was confirmed by analysis of secreted INF-γ in the same culture systems, which showed significantly enhanced production of INF-γ cytokine (**D–F**). The test was done in comparison to tumor cell to T cell ratio without peptide as a negative control sample and PD-1/PD-L1 inhibitor 3 (a cyclic peptide) as a positive control. * *p* < 0.05, ** *p* < 0.01, and *** *p* < 0.001.

**Figure 11 ijms-20-00572-f011:**
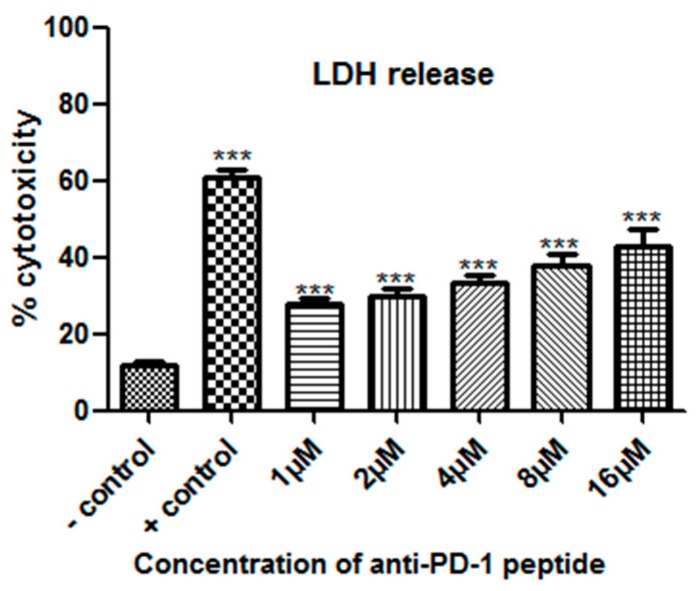
Enhancement of T cell cytotoxicity by FITC-YT-16 incubation. TE-13 cells were co-cultured with effector T cells at a ratio of 16:1 in presence of different FITC-YT-16 concentrations. T cell cytotoxicity was dose-dependently increased with FITC-YT-16 concentration. The highest cytotoxicity was noticed with 16 µM FITC-YT-16. The test was done in comparison to tumor cell to T cell ratio without peptide as a negative control sample and PD-1/PD-L1 inhibitor 3 as a positive control. * *p* < 0.05, ** *p* < 0.01, and *** *p* < 0.001.

**Table 1 ijms-20-00572-t001:** Virtual screening of PD-1 and designed peptides.

Sequences	Scoring
YRCMISYGGADYKCIT(C-C)	−8.675
YRCMISPGGADYKCIT(C-C)	−8.516
YRCMISYGGAEYKCIT(C-C)	−8.502
YRCMISPGGAEYKCIT(C-C)	−8.472
YRCMITYGGGDYKCIT(C-C)	−8.591
YRCMITPGGGDYKCIT(C-C)	−8.472
YRCMITYGGGEYKCIT(C-C)	−8.328
YRCMITPGGGEYKCIT(C-C)	−8.321

**Table 2 ijms-20-00572-t002:** The ratio of target to effector cells.

Ratio	Targeted (Tumor Cells) Number	Effectors Cells (T Cells) Number
64:1	1.28 × 10^7^	2 × 10^5^
32:1	6.4 × 10^6^	2 × 10^5^
16:1	3.2 × 10^6^	2 × 10^5^
8:1	1.6 × 10^6^	2 × 10^5^
4:1	8 × 10^5^	2 × 10^5^

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
