# Peer review of "Design and Synthesis of A PD-1 Binding Peptide and Evaluation of Its Anti-Tumor Activity"

_ijms, 2019, doi:10.3390/ijms20030572_

Round 1

Reviewer 1 Report

The manuscript has been significantly improved compared with the previous version. With the additional control experiments, the work is publishable.

Author Response

A Letter to Reviewer 1

Dear Editor,

Dear reviewer,

Thank you that you have carefully read through the manuscript and gave us so many helpful suggestions. We have carefully read the manuscript and extensively improved the language. The changes to the manuscript were indicated in red color. We also carefully read the comments from the reviewer and tried our best to give proper answers to these comments.

We thank you for your attention to this paper, and look forward to hearing from you at your convenience.

Sincerely yours,

Hanmei Xu and the authors

In responce to the concern of this reviewer, we have the answers in the attached word file. please refer to this file for details.

Reviewer 2 Report

The authors use peptide YT-16 as a PD-1 binding peptide. In general, I find it difficult to know how this peptide compares to other PD-1 binders and what the significance and novelty is to others in this research area. This should be explicitly stated.

I would find it helpful if the authors provided some graphic in the introduction to show t-cell and tumour cell interaction/what specifically they are trying to probe. 

Is cyclisation taken into account in the models? How is the scoring determined? Figure 1 needs to be referenced. Where did YT16 come from?

What are the impurities? Was the compound used for future studies at 90% purity or was it purified and used? 

Is the Kd what was expected? How does this compare to other PD-1 binding moieties? Is there an alternative method to compare the Kd to?

The non-activated t-cell WB sample would be beneficial. Also, the negative controls for PD-1 should be added to the supplementary. 

I would show the separate channels of the zoomed image rather than the zoomed out versions. Please add a scale bar for the zoomed images. What is the concentration of the YT16 and how does it compare to what is observed in the flow cytometry?

What assay was used to determine IL secretion? In the t-cell/tumour cell activity, the tumour cell negative controls are necessary.

Why is the LDH data important? Can you determine an EC50 from the LDH data? Why is the importance of using CD4 and CD8 Tcells? 

The conclusion is quite short. What is the significance of this work, how does it compare to other peptides, is it promising?

Author Response

A Letter to Reviewer 2

Dear Editor,

Dear reviewer,

Thank you that you have carefully read through the manuscript and gave us so many helpful suggestions. We have carefully read the manuscript and extensively improved the language. The changes to the manuscript were indicated in red color. We also carefully read the comments from the reviewer and tried our best to give proper answers to these comments.

We thank you for your attention to this paper, and look forward to hearing from you at your convenience.

Sincerely yours,

Hanmei Xu and the authors

In responce to the reviewer's comments, we drafted a word file and answer the questions one by one. Please refer to the attached word file for details.

Round 2

Reviewer 2 Report

The authors have addressed all of the comments and made significant changes from the initial manuscript. I would ask that they include some of the information from the rebuttal in their manuscript prior to publishing. I think that in the rebuttal the aims of the experiments were much clearer and this should be translated into the manuscript. 

For example:

Activated CD4 T cells can secret cytokines, e.g. IL-2 and IFN-γ, which stimulate the immune cells for proliferation and activation. Therefore, in this experiment we aim to... 

Activated CD8 T cells can recognize tumor cells and kill tumor cells by means of degranulation and Fas/FasL recognition. After tumor cell damage, lactate dehydrogenase (LDH), a cytoplasmic enzyme is released from damaged cells into the supernatant. The levels of LDH provide a measure of the cytotoxic effect of CD8 T cells. Here, we aim to confirm that blocking the interactions between PD-1 and PD-L1 with FITC-YT-16 enhances the cytotoxic effect of CD8 T cells.

By MST assay, YT-16 has a Kd value of 17.8 nM for its target PD-1. Two other peptides are reported to show affinity for PD-1 or PD-L1 [1,2]. The  Ar5Y_4 peptide Kd is 1.38 µM for PD-1 [1] and the DPP-1 peptide Kd is 0.51 µM for PD-L1 [2]. 

Please also include how the cyclisation was taken into account in the screening/docking studies in the methods.

Please make Fig 7 larger, it is quite difficult to read the scale bars.

Fig 5C appears to be stretched. Would it be possible to do all of the graphs in the same format?

Author Response

Answer to reviewer 2

Dear Editor,

Dear Reviewer,

Thank you for sharing time to read the paper and help us to improve the paper. We have read through the paper carefully trying to improve the language. We have edited the Materials and Methods section to more lcearly describe the experimental methods. We also improved the paper according to the reviewer’s suggestions.

We thank you for your attention to this paper, and look forward to hearing from you at your convenience.

Sincerely yours,

Hanmei Xu and the authors

The authors have addressed all of the comments and made significant changes from the initial manuscript. I would ask that they include some of the information from the rebuttal in their manuscript prior to publishing. I think that in the rebuttal the aims of the experiments were much clearer and this should be translated into the manuscript.

For example:

Activated CD4 T cells can secret cytokines, e.g. IL-2 and IFN-γ, which stimulate the immune cells for proliferation and activation. Therefore, in this experiment we aim to...

Answer: we thank the reviewer for this comment. We added a paragraph to the Result section to explain the reason to perform IL-2 and IFN-γ tests.

Activated CD8 T cells can recognize tumor cells and kill tumor cells by means of degranulation and Fas/FasL recognition. After tumor cell damage, lactate dehydrogenase (LDH), a cytoplasmic enzyme is released from damaged cells into the supernatant. The levels of LDH provide a measure of the cytotoxic effect of CD8 T cells. Here, we aim to confirm that blocking the interactions between PD-1 and PD-L1 with FITC-YT-16 enhances the cytotoxic effect of CD8 T cells.

Answer: we thank the reviewer for this comment. We included this part of description in the Result section to explain the reason to perform LDH tests.

By MST assay, YT-16 has a Kd value of 17.8 nM for its target PD-1. Two other peptides are reported to show affinity for PD-1 or PD-L1 [1,2]. The  Ar5Y_4 peptide Kd is 1.38 µM for PD-1 [1] and the DPP-1 peptide Kd is 0.51 µM for PD-L1 [2].

Answer: we thank the reviewer for this comment. We included this part of description in the Discussion section to emphasize that this work is worthwhile to continue.

Please also include how the cyclisation was taken into account in the screening/docking studies in the methods.

Answer: we thank the reviewer for this comment. We added a few sentences in the Result section trying to explain that cyclisation conformation is important for good complementation between PD-1 molecule and peptide YT-16.

Please make Fig 7 larger, it is quite difficult to read the scale bars.

Answer: The Figure is improved according to the reviewer’s suggestion.

Fig 5C appears to be stretched. Would it be possible to do all of the graphs in the same format?

Answer: The Figure is improved according to the reviewer’s suggestion
